# Deciphering the evolution of composite-type GSKIP in mitochondria and Wnt signaling pathways

**Cheng-Yu Tsai**[1,2][�], **Shean-Jaw Chiou**[3][�], **Huey-Jiun Ko**[3,4][�], **Yu-Fan Cheng**[3,4], **Sin-Yi Lin**[3,4], **Yun-Ling Lai**[3,4], **Chen-Yen Lin**[3,4], **Chihuei Wang**[5], **Jiin-Tsuey Cheng**[6], **Hsin-Fu Liu**[7], **Aij-Li Kwan**[1,2,4], **Joon-Khim Loh**[2,4]*, **Yi-Ren Hong**[1,3,4,6]*

1 College of Medicine, Kaohsiung Medical University and National Health Research Institutes, Kaohsiung, Taiwan, 2 Department of Neurosurgery, Kaohsiung Medical University Hospital, Kaohsiung, Taiwan, 3 Department of Biochemistry, College of Medicine, Kaohsiung Medical University, Kaohsiung, Taiwan, 4 Graduate Institute of Medicine, College of Medicine, Kaohsiung Medical University, Kaohsiung, Taiwan, 5 Department of Biotechnology, Kaohsiung Medical University, Kaohsiung, Taiwan, 6 Department of Biological Sciences, National Sun Yat-Sen University, Kaohsiung, Taiwan, 7 Department of Medical Research, Mackay Memorial Hospital, Taipei, Taiwan

☉ These authors contributed equally to this work.
* m835016@kmu.edu.tw (YRH); jokhlo@kmu.edu.tw (JKL)

**Data Availability Statement:** All relevant data are within the paper and its Supporting Information files.

## Abstract

We previously revealed the origin of mammalian simple-type glycogen synthase kinase interaction protein (GSKIP), which served as a scavenger and a competitor in the Wnt signaling pathway during evolution. In this study, we investigated the conserved and nonconserved regions of the composite-type GSKIP by utilizing bioinformatics tools, site-directed mutagenesis, and yeast two-hybrid methods. The regions were denoted as the pre-GSK3β binding site, which is located at the front of GSK3β-binding sites. Our data demonstrated that clustered mitochondria protein 1 (CLU1), a type of composite-type GSKIP that exists in the mitochondria of all eukaryotic organisms, possesses the protein known as domain of unknown function 727 (DUF727), with a pre-GSK3β-binding site and a mutant GSK3β-binding flanking region. Another type of composite-type GSKIP, armadillo repeat containing 4 (ARMC4), which is known for cilium movement in vertebrates, contains an unintegrated DUF727 flanking region with a pre-GSK3β-binding site (115SPxF118) only. In addition, the sequence of the GSK3β-binding site in CLU1 revealed that Q126L and V130L were not conserved, differing from the ideal GSK3β-binding sequence of simple-type GSKIP. We further illustrated two exceptions, namely 70 kilodalton heat shock proteins (Hsp70/DnaK) and Mitofilin in nematodes, that presented an unexpected ideal GSK3β-binding region with a pre-GSK3β sequence; this composite-type GSKIP could only occur in vertebrate species. Furthermore, we revealed the importance of the pre-GSK3β-binding site (118F or 118Y) and various mutant GSK3β-binding sites of composite-type GSKIP. Collectively, our data suggest that the new composite-type GSKIP starts with a DUF727 domain followed by a pre-GSK3β-binding site, with the subsequent addition of the GSK3β-binding site, which plays vital roles for CLU1, Mitofilin, and ARMC4 in mitochondria and Wnt signaling pathways during evolution.

**Funding:** This work was supported by Kaohsiung Medical University Hospital, Taiwan, KMUH109-9R28 (awarded to JKL), KMUH106-M18 (awarded to CYT); NSYSU-KMU Joint Research Project, Taiwan NSYSUKMU-109-P006 and NSYSUKMU-110-P009 (awarded to YRH and JTC); Ministry of Science and Technology, Taiwan MOST109-2320-B037-032; 109-2320-B037-012; 110-2320-B037-029 (awarded to YRH)."

**Competing interests:** The authors have declared that no competing interests exist.

## Introduction

Glycogen synthase kinase interaction protein (GSKIP) was originally cloned and identified as a glycogen synthase kinase-3β (GSK3β) interacting protein through yeast two-hybrid screening using GSK3β [1–4]. Subsequently, GSKIP was also characterized as a small A-kinase anchoring protein (AKAP) [5, 6]. Studies have suggested that GSKIP is a cytosolic scaffolding protein that contains protein kinase A (PKA) RII- binding sites at the N-terminal residue V41/L45 and a GSK3β-binding domain at the C-terminal residue L130 [1–3, 5, 6].

In a previous study, transfection of the neuronal-like SH-SY5Y cell line with wild-type GSKIP inhibited neurite outgrowth, implying the possible role of GSKIP in the neuronal system [7]. GSKIP was demonstrated to modulate dynamin-related protein 1 (Drp1) phosphorylation and to provide neuroprotection against oxidative stress for SH-SY5Y cells [8]. These findings indicate that GSKIP functions as an anchor protein for GSK3β in the cAMP/PKA/Drp1 signaling axis. Moreover, it was reported that GSKIP could interact with PKA and GSK3β, which is required for Wnt signaling regulation, via a cytoplasmic destruction complex with β-catenin, leading to the negative regulation of Wnt signaling [6, 9]. Using knockout animal models, Deák et al. demonstrated that GSKIP regulates GSK3β activity and controls palatal shelf fusion during mice embryonic development [10]. GSKIP is associated with a predisposition to familial myeloid malignancies and poses a challenge for hematologists in germ-line gene duplication studies [11–13]. GSKIP is also involved in the PPARγ-related angiogenic potential of endothelial pulmonary microvascular endothelial cells [14]. A recent study showed that miR-150-5p significantly inhibited Wnt-β-catenin signaling by simultaneously targeting GSKIP and β-catenin in non-small-cell lung cancer (NSCLC) cells [15]. GSKIP was downregulated in disease-related cellular protein networks under different epidermal growth factor receptor (EGFR) mutations in NSCLC [16]. Moreover, long intergenic nonprotein coding RNA 173 (Linc00173) upregulated Etk by functioning as a competitive endogenous RNA "sponging" miRNA-218, leading to GSKIP upregulation and resulting in the translocation of β-catenin in small-cell lung cancer cells [17]. Additionally, miR-181c-5p was reported to mitigate the cancer cell characteristics and invasive properties of cervical squamous cell carcinoma by targeting the GSKIP gene [18]. In a previous study, we demonstrated that this protein PKA/GSKIP/GSK3β axis plays a role in Tau phosphorylation in Alzheimer's disease [19]. Collectively, these findings indicate that (1) simple-type GSKIP may play a multifaceted role in several diseases in a context-dependent manner, and (2) it is involved in the regulation of mitochondria proteins and the Wnt signaling pathway.

GSKIP has two subtypes, simple and composite, which coexist in all prokaryotic and eukaryotic cells. In our previous study, we revealed the origin of mammalian simple-type GSKIP and its evolution and discovered that it is involved in the Wnt signaling pathway not only as a scavenger but also as a competitor [20]. However, the origin and evolution of composite-type GSKIP remain unknown. In the present study, we generated evidence to demonstrate that along with simple-type GSKIP-linked proteins, four groups of composite-type GSKIP, namely clustered mitochondria protein 1 (CLU1) and Mitofilin in mitochondria, heat shock protein 70 (Hsp70) chaperone, and Armadillo repeat-containing 4 (ARMC4), are involved in Wnt signaling. This can be used to uncover the origin of the GSK3-binding site during the evolution of the protein known as domain of unknown function 727 (DUF727) and four groups of composite-type GSKIP.

## Materials and methods

### Gene ontology and UniProt analysis

Gene Ontology (http://geneontology.org) and InterPro (http://www.ebi.ac.uk/interpro/) were used for data mining [21–23]. UniProt (https://doi.org/10.1093/nar/gkaa1100) was combined

with InterPro [21–25] to compare the results available at the National Center for Biotechnology Information (NCBI) website for domain enrichment by using the E-Value for parameters. Gene-based tests were performed using the following keywords as indicators: GSKIP, DUF727, GSK3β, CLU1, and AMRC4 separately and in combination with GSKIP-related data (S1 Fig). We used FlyBase version FB2014_04 to identify conservative portions in InterPro. Panther data mining is a protein classification system that is applied for analysis based on evolutionary relationships. It is a large biology database of gene/protein families and their functionally related subfamilies. It can be used to classify and identify the function of gene products [26, 27].

## Phylogenetic analysis

We used ClustalW [28] for multiple GSKIP sequence alignment for DUF727 (50–100 amino acids), pre-GSK3β (115–118 amino acids), and the GSK3β-binding domain (122–130 amino acids). The maximum likelihood method was used to select CLU1 to generate an unrooted phylogenetic tree of 20 species; a phylogenetic tree of 18 species was also found for one bacterium and ARMC4 orthologs [29–31]. ClustalW was used to ensure the tree structure [28]. The accession numbers of SSF103107, PF05303, IPR007967, and IPR023231 of all species in the UniProt search (S1 Table) and the sequence alignment and phylogenetic analysis of CLU1 and ARMC4 (S1 Fig) were used as parameters. The T-Coffee method was used to align full-length CLU1 and ARMC4 proteins and their fragmentary proteins spanning the SSF103107 unintegrated superfamily [32–34]. The MEGA-X program was used to produce the best-fitting amino acid substitution model [29–31]. We used the maximum likelihood method from MEGA-X to reconstruct phylogenetic trees with the LG substitution model [35, 36]. Boot strapping was used to evaluate the robustness of the phylogenetic trees.

## Molecular modeling comparison

The initial three-dimensional (3D) NMR structural model of GSKIP (PDB ID: 1SGO) was obtained as previously described [20]. The NorthEast Structural Genomics consortium was used to obtain the human NMR structure of GSKIP (C14orf129, HSPC210; PDB ID: 1SGO). Next, the 3D structure of the ACT domain, which folds with a ferredoxin-like βαββαβ topology [37], was determined. The structure was minimized for 100,000 conjugate gradient steps and then subjected to 100-s isothermal, constant-volume MD simulation. The final structure was used in domain comparisons (GSKIP, ACT domain, and CLUH-KIAA0664-SSF103107) [1–4, 6, 7, 37].

## Consensus sequences and sequence logos: Weblogo 3.6.0

To illustrate the consensus sequence logos of GSKIP, we presented typical simple-type logos and several composite types. The rules were adapted from those used by the sequence comparison website PROSITE (http://prosite.expasy.org/sequence_logo.html) [38], and standard one-letter codes were used for amino acids.

## NetWheels: Helical wheel diagram of peptides and net projections maker

The helices found in peptides and proteins are commonly modeled in two dimensions [39]. They can offer a view of the central axis in a protein. Wheel and net projections have been used to represent the two dimensions of 3D helical structures, and they enable the observation of the helical structural properties, especially in terms of residue polarity and intramolecular bonding. We used the helical wheel diagram shown here to determine the distribution of

amino acid residues in a helical segment within the sequence of simple-type GSKIP and various composite-type variants to distinguish the differences between the two types.

## Cloning, site-directed mutagenesis, and DNA sequencing

The plasmids of pACT2-GSKIP and pAS2-1-GSK3β were constructed for the yeast two-hybrid assay as described previously [40–42]. Briefly, GSK3β was cloned in-frame with the Gal4 DNA-binding domain in the pAS2-1 vector (MATCHMAKER Two-HybridSystem 2, Clontech) to yield the pAS2-1-GSK3β bait plasmid. In addition, DNA fragments encoding GSKIP were amplified through PCR using Taq polymerase (TaKaRa). The PCR fragments were then inserted into the BamHI and XhoI sites of the pACT2 (Clontech) vector to construct the pACT2-GSKIP plasmid. GSKIP Y118P, Y118A, Y118F, F122P, F122A, F122Y, L126P, F126A, L126V, L126Q, L130P, L130A, L130I, and L130V mutants were created through a site-directed mutagenesis technique by using the QuikChange Lightning kit (GE Healthcare, Sunnyvale, CA, USA). Mutated nucleotides were verified using an ABI PRISM 3730 Genetic Analyzer (Perkin-Elmer) for DNA sequencing. All experimental procedures were performed in accordance with the manufacturer's protocol.

## Yeast two-hybrid system

Yeast two-hybrid screening was performed using the MATCHMAKER Two-Hybrid System 2 (Clontech) [40–42]. YRG-2 yeast host cells were purchased from Stratagene. pAS2-1 and pACT2 plasmids were cotransfected and selected on G2 plates deficient in tryptophan and leucine and on G3 plates deficient in histidine. The yeast host cells were MATa ura3–52 his3–200 ade2–101 lys2–801 trp1–901 leu2–3 112 gal4–542 gal80–538 LYS2::UASGAL1-TATA GAL1-- HIS3 URA3::UASGAL4 17mers(x3)-TATACYC1-lacZ. A visible blue-color pattern in the colony filter lift assay on the G3 plates represented a positive interaction [41]. YRG-2 yeast cells were cotransfected with pACT2-GSKIP and an empty pAS2-1 vector and spread on G2 and G3 agar plates to determine the growth-inhibiting effect of GSKIP in yeast.

## Results

### Analysis of UniProt and GSKIP gene ontology: Identification of composite-type GSKIP

UniProt was used in combination with InterPro for domain enrichment analyses. The results were compared with the results in the NCBI database through stringent and extended gene-based tests for ranking the genes (Fig 1A, 1B and 1C). Our survey revealed that several domains were enriched for protein entries. The evolution of simple-type GSKIP and the GSK3β- and PKA RII-binding domains of 52 species has already been clearly demonstrated in previous reports [9, 20]. In this study, we extended the findings concerning four composite-type GSKIPs, namely CLU1 and ARMC4 (through the gene fusion mechanism presented in Fig 1A, 1B and 1C as well as the next subsection below) and another two proteins with sporadic occurrence (Mitofilin and Hsp70/Dnak), in nematodes (see the succeeding section on the hijack mechanism). CLU1 and ARMC4 contain DUF727; however, CLU1 contains other parts of CLU-N, CLU, CLU central, winged helix-like DNA-binding domains, and tetratricopeptide-like helical domains, but ARMC4 only contains 13X Armadillo repeats [43, 44]. DUF refers to a "domain of unknown function" [19–23, 45]. A short domain in the clustered mitochondria protein is involved in its mitochondrial cytoplasmic distribution [46, 47]. Moreover, ARMC4 in vertebrates has been identified as a multiprotein complex responsible for cilia movement; it is necessary for targeting and anchoring outer dynein arms [43, 44]. In the

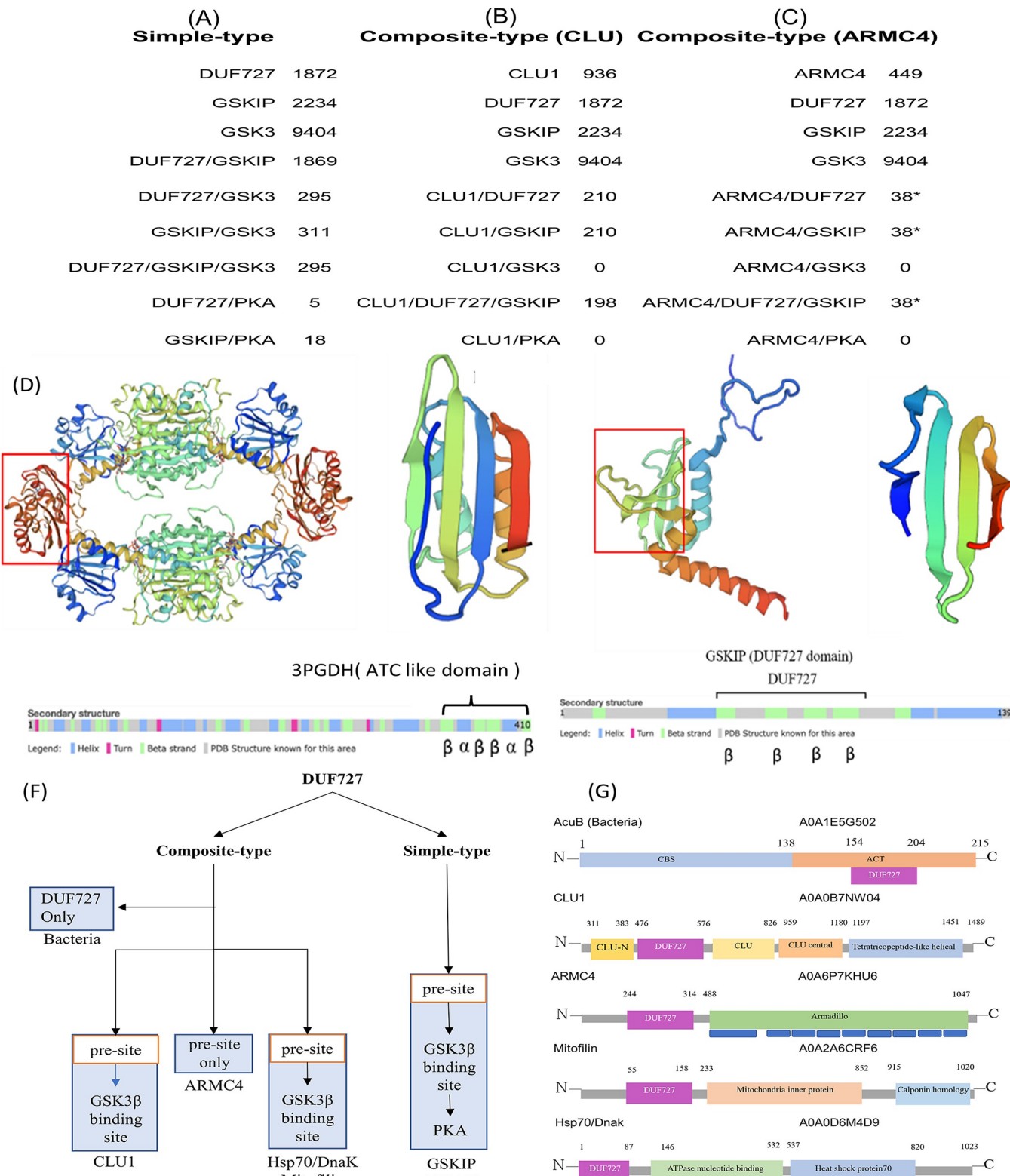

**Fig 1. Summary of UniProt and InterPro data mining to reveal two types of GSKIP.** A, Evolved simple-type GSKIP from DUF727 to pre-GSK3β and then evolution to the GSK3β-binding site. The results of UniProtKB data retrieval using keywords are provided. When using GSKIP as a keyword, a total of 2234 protein entries were found, but the records were reduced when other terms such as DUF727, GSK3β, and PKA were added. B, When using CLU1 as a keyword, 936 protein entries were found, and 210 protein entries were convergent with the keywords "DUF727" or "GSKIP." C, Using ARMC4 as a keyword resulted in

449 protein entries, but only 38 records remained as the SSF103107 hidden code (marked with "*" as a cryptic code; see the text for details) when DUF727 or GSKIP was combined with AMRC4 for data mining. D, *E. coli* 3PGDH with a pair of ACT domains formed an eight-stranded antiparallel sheet whose 3D structure was determined (left) to fold into a ferredoxin-like βαββαβ topology (red box, right panel). E, NMR analysis of DUF727 (PDB ID: 1SGO) with the central part of antiparallel ββββ topology (red box, right panel) based on the Northeast Structural Genomics consortium (left). F, All queries of previous simple-type GSKIP were summarized using UniProt and Panther data mining (see the text for details). G, Four groups of composite-type GSKIP: AcuB, CLU1, ARMC4, and sporadic (Hsp70/Dnak and Mitofilin).

present study, data mining was performed to search the available protein entries, with the results yielding 210 of 936 (22%) with respect to CLU1, 198 of 936 (21%) with respect to CLU1/DUF727, and 210 of 936 (22%) with respect to CLU1/GSKIP. Additionally, the hidden (cryptic) superfamily of DUF727 (SSF103107) was indicated to be an unintegrated family. The hypothetical protein c14orf129 hspc210 superfamily entry could enrich entries for the GSKIP (SSF103107) domain to 1869/1872 (99.8%; Fig 1A, 4 vs. 1). When ARMC4 was retrieved (all entries must be searched to find the hidden code SSF103107), only the hidden (cryptic) super-family of DUF727 (identified as the unintegrated family SSF103107) with 38/449 entries (8%) was shown to contain a pre-GSK3-binding site (115SPxF118) containing F118 instead of Y118 as compared with the normal GSK3-binding site (more details on the comparisons are provided in subsequent discussion). We occasionally found composite-type GSKIP in two bacterial species, one of which was gram-positive *Desulfuribacillus alkaliarsenatis* (A0A1E5G502_9BACL). The acetoin-utilization protein AcuB contained the CBS and ACT domains (CBS, 1–138 aa; ACT, 139–212 aa; GSKIP; DUF727, 154–204 aa), and DUF727 was located within the ACT domain as composite GSKIP. The ACT domain was identified in a PSI-BLAST search. *Escherichia coli* 3PGDH was discovered to be the first protein with an ACT domain that folds into a ferredoxin-like βαββαβ topology [37] (Fig 1D). The ACT domain is found in a variety of contexts and may be a conserved regulatory ligand-binding fold. However, DUF727 has antiparallel ββββ topology (PDB ID: 1SGO, the NorthEast Structural Genomics consortium was used for NMR analysis; Fig 1E), with a similar β-sheet homology (ferredoxin-like βαββαβ topology; Fig 1D right panel compared with Fig 1E) as the human CLUH-KIAA0664-SSF103107 superfamily domain (Fig 1E). We propose that during evolution, GSKIP may have originated from DUF727, as found in bacteria, and may have acquired the pre-GSK3β-binding motif and GSK3β-binding domain to become the ancestor of simple-type GSKIP. Through evolution, simple-type GSKIP later acquired the PKA RII-binding domain as GSKIP/AKAP in vertebrates, whereas simple-type GSKIP has been retained in invertebrates. For composite-type GSKIP, one bacterium was found to contain DUF727 with the ACT domain, whereas in all eukaryotes, DUF727 was incorrectly recognized as GSKIP. Although DUF727 lacks the ideal GSK3β-binding domain, it was still counted among the CLU/TIF31 and CLUH/KIAA0664 proteins. In some vertebrates, the GSKIP domain is also found in an ARMC4-containing protein with a pre-GSK3β-binding site only. Two sporadic Mitofilin and Hsp/DnaK proteins were found in invertebrates containing perfect pre-GSK3β-binding and GSK3β-binding sites (Fig 1F). We summarize four groups of composite-type GSKIP in Fig 1G.

## Two groups of composite-type GSKIP families may have evolved through a gene-fusion mechanism: CLU1 (mitochondria) and ARMC4 (Wnt pathway)

In different species, to determine the conserved sequences of GSK3β-binding regions and DUF727 in CLU1, we first selected one bacterial prototype and 20 GSKIP orthologs and used ClustalW and MEGA-X to construct a phylogenetic tree (Figs 2 and S1A). These 20 species

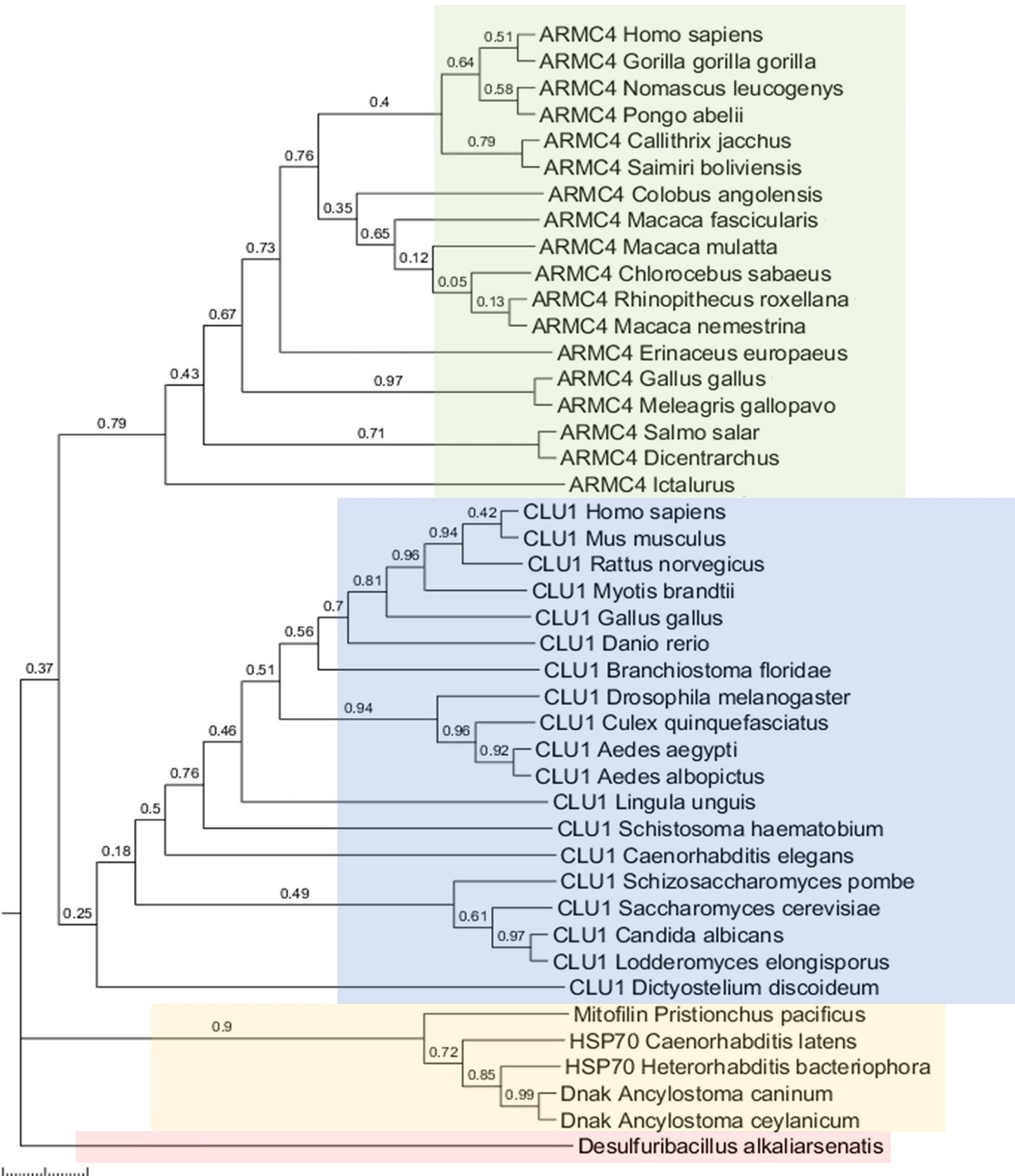

**Fig 2. Phylogenetic analysis of four composite-type GSKIP proteins, namely ARMC4, CLU1, HSP70/Dnak and Mitofilin, among various species.** The evolutionary history of GSKIP was inferred using the maximum likelihood method. Initial trees for the heuristic search were obtained automatically by applying the neighbor-joining and BioNJ algorithms to a matrix of pairwise distances estimated using the JTT model and by then selecting the topology with the superior log likelihood value. The tree is drawn to scale, with branch lengths measured in the number of substitutions per site. This analysis involved 43

amino acid sequences. The final dataset included a total of 161 positions. Phylogenetic tree generated using MEGA-X [31]. The bootstrap values represented as the likelihood function for each species as indicated.

comprised one bacteria [*D. alkaliarsenatis* (WP_069641749.1)] (prototype), four yeasts [*Saccharomyces cerevisiae* (NP_013725.1), *Schizosaccharomyces pombe* (NP_595319.2), *Candida albicans* (XP_710862.2), and *Lodderomyces elongisporus* (XP_001526951.1)], one slime mold [*Dictyostelium discoideum* (XP_629429.1)], one flatworm [*Schistosoma Haematobium* (XP_012796625.1)], one Nematoda [*Caenorhabditis elegans* (NP_499097.1)], four insects [*Drosophila melanogaster* (NP_646986) and *Culex quinquefasciatus* (XP_001843004.1)], including two species of mosquitoes [*Aedes aegypti* (XP_001651007.1) and *Aedes albopictus* (A0A182G693)], one brachiopod [*Lingula anatina* (XP_013383658.1)], and seven Chordata [*Branchiostoma lanceolatum* (XP_002595966.1), *Myotis brandtii* (Brandt's bat, S7P578), *Danio rerio* (E7FE02), *Gallus gallus* (XP_015151355.1), *Mus musculus* (NP_001074627.1), *Rattus norvegicus* (NP_001292142.1), and *Homo sapiens* (NP_056044.3)]. Additionally, ClustalW was combined with the T-Coffee web-based program and MEGA-X to ensure a tree structure (Fig 3). Both a GSK3β-binding motif (122Fxxx126LxxR/K/QL130) and a pre-GSK3-binding motif (115SPxF118 rather than 115SPxY118 in simple-type GSKIP) were found in most of the GSKIP orthologs (Fig 3).

We found that two distinct bacteria contained composite-type GSKIP based on the observation of the aligned sequences without the GSK3β-binding site, which revealed that they are primitive [5, 6, 20]. Why and how composite-type GSKIP evolved a pre-GSK3β-binding site prior to the GSK3β functional domain should be investigated, and such an investigation could provide some indication of the role of the GSK3β-binding site in composite-type GSKIP evolution.

When DUF727 is regarded as an initial domain, the CLU1 mitochondria family [48, 49] and the ARMC4 family of proteins that emerge [43, 44] are composite-type GSKIPs. By contrast, in simple-type GSKIP, the GSK3β-binding region is conserved with some residues, indicating that the 122Fxxx126L/QxxR/KV/G/A130 region is essential. The Leu130 residue has been characterized as being essential for GSK3β binding in humans [1, 6, 20], indicating its inactivation when this site is modified with GSK3β. In the present study, this always occurred in CLU1 (Table 1), indicating that CLU1 is still in the process of evolution. Of note, the modification of the ARMC4-pre-GSK3 site 118F to 118Y abolished its interaction with GSK3 (Table 1), indicating the same evolutionary process of the ARMC4-pre-GSK3β site as that of the CLU1 family. When using the helical wheel diagram, the comparisons revealed various V126L residues, but no 130L residue, in CLU1 (Fig 4, GSKIP: 122–139 aa with its various mutants) and Logo 3.6.0 (Fig 5A–5D). These data enabled the evaluation of whether the consensus sequence is conserved in pre-GSK3- (115SPxF/Y118; Fig 5B and 5C compared with Fig 5A) and GSK3β-binding sites (122Lxxx126LxxK/RL130, Fig 5D compared with Fig 5A). Crucially, three composite GSKIP variants (126Q, 126V, and 130V) from different species were found to bind GSK3 by Y2H (Table 1). In addition, the DUF727 domain seemed to gradually become a pre-GSK3β-binding site (115SPxF118) as a flanking region with the consensus sequence site 115SPxF118xxx122Fxxx126LxxR/KL130, whereas in vertebrates, ARMC4 contained a hidden code SSF103107 (unintegrated DUF727 superfamily, Fig 5C) flanking region with pre-GSK3β-binding sites at 115SPxF118 only (see the next subsection below).

## A hidden code SSF103107 domain as a metaphor of DUF727 in the ARMC4 family

In our previous study, we used the keywords DUF727 (PF05303) and GSKIP (IPR007967) to search for ARMC4. Subsequently, DUF727 and GSKIP were eliminated as keywords from the

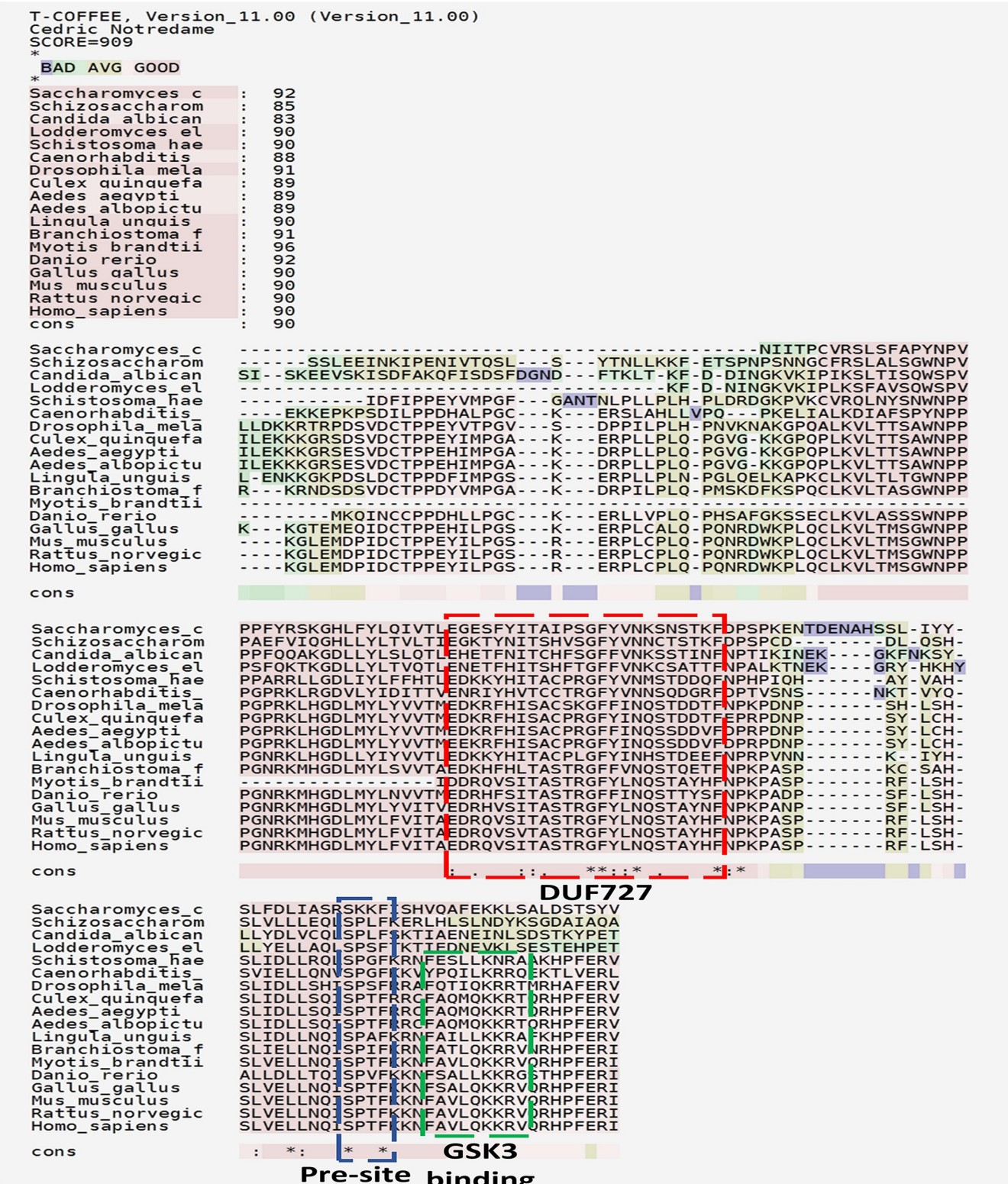

**Fig 3. Sequence alignment of 20 composite-type GSKIP/CLU1 proteins among various species.** Multiple sequence alignment with respect to the DUF727 region of GSKIP orthologs using T-Coffee. The conserved residues are indicated with asterisks, and residues with high similarity among the orthologs are marked with dots at the bottom. The 115SPxF118xxx122FxxxQxxRV130 motif in the pre-GSK3β- and GSK3β-binding regions is also indicated by blue and green boxes, respectively. T-Coffee also estimates alignment accuracy and improves phylogenetic tree reconstruction.

**Table 1. Mutagenesis assay of the pre-GSK3β-binding site (118F or 118Y) and various mutant GSK3β-binding sites of full-length GSKIP and GSKIPtide (115–139).**

| GSKIP | | | GSKIPtide | | |
|---|---|---|---|---|---|
| | G2 | G3 | | G2 | G3 |
| WT | + | + | WT | +++ | +++ |
| Y118P | +++ | - | Y118P | +++ | - |
| Y118A | +++ | +++ | Y118A | +++ | - |
| Y118F | +++ | +++ | Y118F | +++ | -* |
| F122P | +++ | - | F122P | +++ | - |
| F122A | +++ | +++ | F122A | +++ | - |
| F122Y | +++ | +++ | F122Y | +++ | - |
| L126P | +++ | - | L126P | +++ | - |
| L126A | +++ | +++ | L126A | +++ | - |
| L126V | +++ | +++ | L126V | +++ | - |
| L126Q | +++ | +++ | L126Q | +++ | - |
| L130P | +++ | - | L130P | +++ | - |
| L130A | +++ | +++ | L130A | +++ | - |
| L130I | +++ | +++ | L130I | +++ | - |
| L130V | +++ | +++ | L130V | +++ | - |

The β-galactosidase filter assay was also used for semiquantitative analysis. (+) indicates positive interaction, (−) indicates the interaction is abolished, and (+++) indicates a strong interaction.

The red lines indicate the pre-GSK3β-binding site and various GSK3β-binding sites.

* Data are comparable to those of Hundsrucker et al., 2010 [6].

website. Only the hidden code SSF103107 was used as a metaphor for DUF727 and GSKIP for searching for ARMC4. Although the hidden code has been classified into the DUF727 super-family, the hidden code SSF103107 domain shares homology with DUF727 (PF05303) and GSKIP (IPR007967). Therefore, we checked all ARMC4 family queries, and 38 of 449 species (8%) were found to contain the hidden code SSF103107, with protein evolution through gene (domain) fusion. Because the pre-GSK3β-binding site (115SPxF118) was retained only in the ARMC4 family in vertebrates, the GSK3β-binding site is not yet present at this stage. The pre-GSK3β-binding site (115SPxF118) may exist as a primitive type in vertebrates. We selected 18 SSF103107 orthologs from vertebrates to determine their alignment and performed T-Coffee phylogenetic tree construction of ARMC4 proteins evolved through gene fusion in 8% of vertebrate species (S1B Fig). We selected SSF103107 orthologs, and ClustalW and MEGA-X were used for phylogenetic tree construction (S1C Fig). Proteins encoded by ARMC4, 10 armadillo repeat motifs (ARMs), and one HEAT repeat were found in ARMC4, which is thought to be involved in ciliary and flagellar movement [43, 44]. The term armadillo is derived from the historical name of the β-catenin gene in the fruit fly *Drosophila*, where the armadillo repeat was first discovered. Although β-catenin was previously thought to be a protein involved in linking cadherin cell adhesion proteins to the cytoskeleton, a recent study indicated that β-catenin regulates the homodimerization of alpha-catenin, which in turn controls act branching and bundling [50]. However, the armadillo repeat has been found in a wide range of proteins with other functions. This protein domain plays a vital role in transducing Wnt signals during embryonic development [51].

As described in the earlier text, we found that a substantial number of DUF727 domain insertions into CLU1 (in mitochondria) together with ARMC4 (in the Wnt pathway) may constitute the gene fusion recombination mechanism. We also detected ARMC4 in the pre-

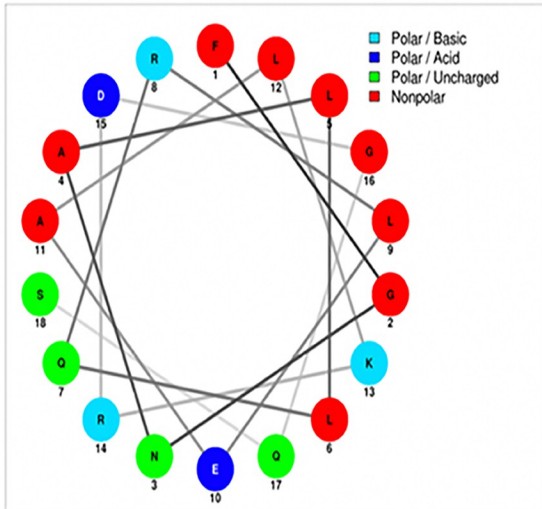

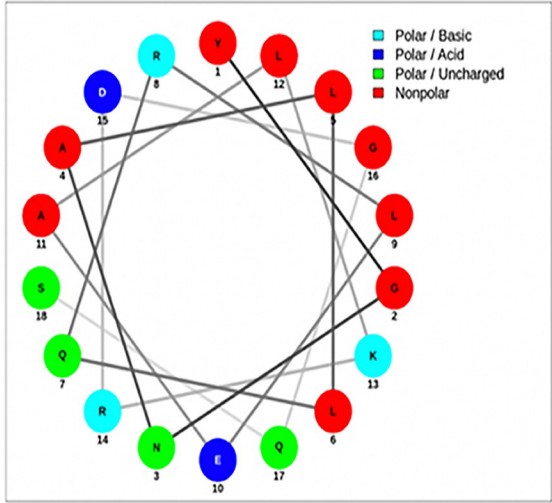

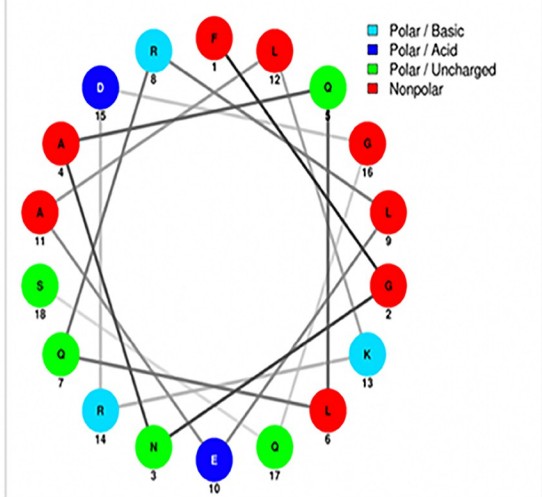

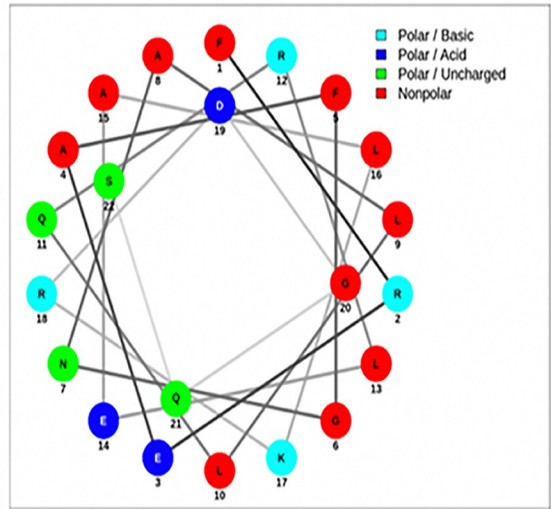

**Fig 4. Helical wheel analysis of GSKIP wt and various GSKIP mutants at amino acid sequence 122–139, which harbors the GSK3β-binding site.** A, GSKIP wt. B, GSKIP L126Q mutant. C, GSKIP F122Y mutant. D, GSKIP Y118F mutant.

GSK3β-binding site (115SPxF118, Fig 5C), whereas the CLU1 family gradually formed from the pre-GSK3β-binding site to extend GSK3β-binding sites (115SPxF118xxxF122xxxQ/V126xxRV130, Fig 5B). However, no PKA-binding sites were found in composite-type GSKIP compared with simple-type GSKIP. We discovered evidence that the contribution of domain fusion to the evolution of multidomain proteins is bounded by the lower boundary of 63% in invertebrates and the upper boundary of 94% in vertebrates in the CLU1 family (Fig 3).

By contrast, in the vertebrate ARMC4 family, a cryptic (unintegrated) superfamily DUF727 (hidden code SSF103107) was also found to bind to the DUF727 domain in 8% of species with

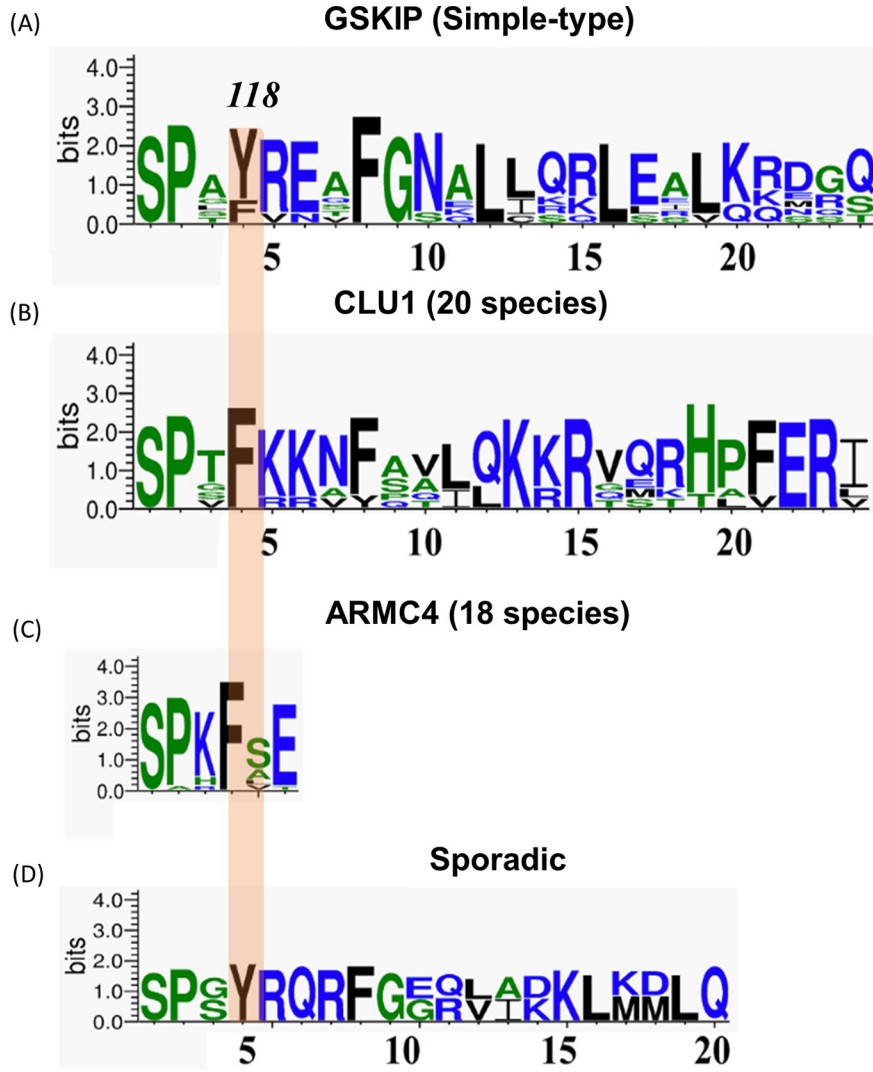

Representation of simple-type (A) and Composited-type Consensus sequences of GSKIP (B,C,D)

(A): $S^{115}Px[YF]^{118}xxxL^{126}xx[RKQ]^{129}L^{130}xx[LV]^{133}$

(B): $S^{115}PxF^{118}xxx[FY]^{122}xxx[QL]^{126}xxR^{129}[VGQT]^{130}xx[HT]^{133}$

(C): $S^{115}PxF^{118}xE^{120}xM^{122}$

(D): $S^{115}PxY^{118}xQxF^{122}xxx[LV]^{126}xxK^{129}L^{130}xxL^{133}$

[YF] means Tyr or phe; [RKQ] means Arg or Lys or Gln; x indicates any amino acid; ARMC4 contains pre-GSK3β binding sites in 20 species are also shown

**Fig 5. Differences in pre-GSK-binding site sequences.** Four consensus GSKIP sequences were represented using Weblogo 3.6.0. A, GSKIP (simple-type). B, CLU1 (CLUH). C, ARMC4. D, Sporadic (Mitofilin and Hsp70/Dnak).

a pre-GSK3β-binding site (115SPxF118) (20 species compared with one bacterium). The association of DUF727 with the pre-GSK3β-binding site in vertebrate ARMC4 suggests that DUF727 (SSF103107) originates in a primitive stage of evolution. We suggest that the gene

fusion mechanism is a major contributor to the evolution of the CLU1 (in mitochondria) and ARMC4 (in the Wnt pathway) families in composite-type GSKIP.

## Hijack (recombination) mechanism: A noncanonical order (strange order) of Hsp70/DnaK and Mitofilin in evolution

Members of the Hsp70 family are ubiquitously expressed and highly conserved; for example, the major form of Hsp70 from *E. coli*, termed DnaK, is approximately 50% identical to human Hsp70s. Hsp70 chaperone-assisted folding involves repeated cycles of substrate binding and release. Hsp70 activity is ATP-dependent. Hsp70 proteins comprise two regions: the amino terminus, which is the ATPase domain, and the carboxyl terminus, which is the substrate-binding region [52, 53].

Unexpectedly, in this study, we demonstrated the noncanonical order of the sporadic Hsp70/DnaK (A0A261CFR1_9PELO, *C. latens*; A0A1I7XTM9_HETBA, *Heterorhabditis bacteriophora*; A0A368GFB0_ANCCA, *Ancylostoma caninum*; and A0A0D6M4D9_9BILA, *A. ceylanicum*) and Mitofilin (H3EKR7_PRIPA, *Pristionchus pacificus*) proteins found in the composite-type GSKIP of invertebrate nematodes; they exist in the mature forms 115SPxY118xxxFxxxLxxRL130 and 115SPxY118xxxFxxxV116xxKL130, respectively (Fig 5D). The pre-GSK3-binding site prefers 118Y instead of 118F in invertebrate nematodes compared with the CLU1 (all eukaryotes) and ARMC4 (vertebrate) families. These pre-GSK3β-binding sites utilizing 118Y instead of 118F to exhibit their GSK3β-binding activity were completely conserved in Mitofilin and Hsp70/DnaK in invertebrate nematodes but not vertebrates. The noncanonical order of these two composite-type GSKIPs in the evolutionary tree could be explained by a hijack (recombination) mechanism, as further evidenced by omega and T-Coffee comparisons. Apparently, the DUF727 domain gradually evolved as a pre-GSK3β-binding site (115SPxF118) flanking region with the consensus sequence site 115SPxF118xxx122FxxxLxxR/KL130 (Fig 5A), whereas in vertebrates, ARMC4 only contained the cryptic SSF103107 (unintegrated DUF727 superfamily, Fig 5C) flanking region with the pre-GSK3β-binding sites 115SPxF118 lacking the GSK3β-binding region (122FxxxLxxR/KL130). Altogether, these findings suggest that (1) at the pre-GSK3β-binding site, 118F is present prior to 118Y during evolution; (2) ARMC4 in higher organisms evolved through the GSK3-binding site of Hsp70/DnaK and Mitofilin; and (3) composite-type GSKIP evolved slower than simple-type GSKIP. We further performed site-directed mutagenesis and yeast two-hybrid assays to compare and reveal the importance of this pre-GSK3β-binding site with its flanking GSK3β-binding conserved sites in invertebrates and vertebrates (Table 1).

## Human pre-GSK3β- and GSK3β-binding sites of the GSKIP domain in a budding yeast (*S. cerevisiae*) model

We used budding yeast (*S. cerevisiae*) as an ideal model organism for studying domains with pre-GSK3β- and GSK3β-binding regions to determine how a new composite-type GSKIP was conserved in multicellular organisms through natural selection and it did not interfere with endogenous simple-type GSKIPs. As described in the earlier text, the pre-GSK3β-binding site (118F or 118Y) and various mutant GSK3β-binding sites (from CLU1) were compared through site-directed mutagenesis, as in a previous study, using yeast two-hybrid methods. Our data revealed 118Y, 122F, 126F, and 130L to be essential for binding during evolution, which is consistent with our previous study results [20]. Taken together, these data imply that the pre-GSK3β-binding site with 118Y plays a crucial role for GSK3β-binding sites as well as 122F, 126L, and 130L (Table 1) [1, 3, 20].

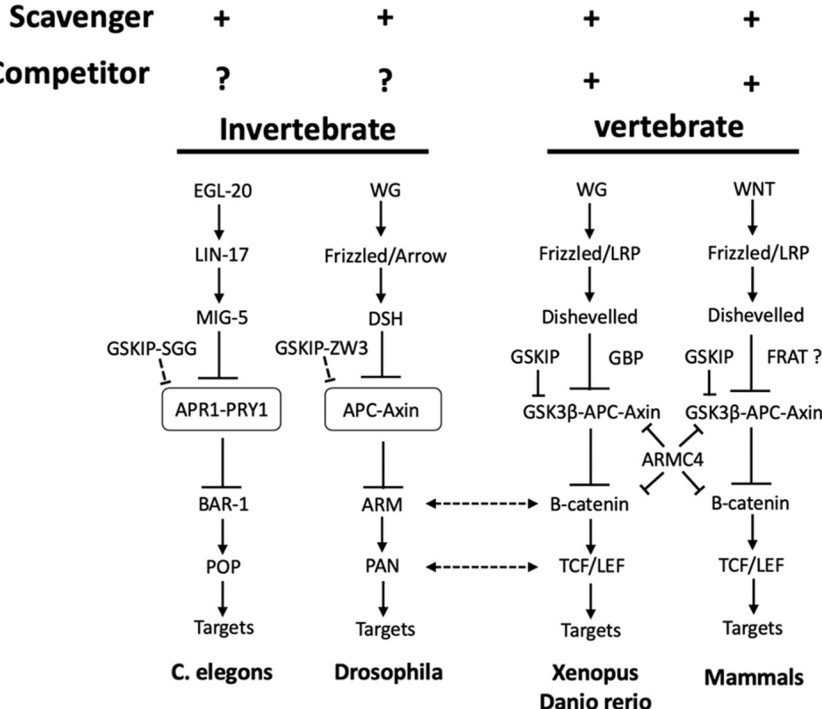

**Fig 6. Physiological implications and importance of both simple-type and composite-type GSKIP in Wnt signaling.**
GSKIP may play different roles in the Wnt pathway in different species. For *C. elegans* and *Drosophila*, due to a lack of the Axin GSK3β-binding motif on either the APR1-PRY1 or APC-Axin complex, GSKIP orthologs (GSKIP-SGG and GSKIP-ZW3) may serve only as scavengers that prevent complex formation. For vertebrates such as *Xenopus*, *D. rerio*, and mammals, GSK3β is one of the proteins involved in the formation of the destruction complex with β-catenin. Thus, GSKIP as a negative regulator may play a dual role as a scavenger to prevent GSK3β binding to the destruction complex or as a competitor for the Axin GSK3β-binding site with GBP or FRAT involved in the canonical Wnt pathway. ARMC4 may function as a competitor to APC and β-catenin. In addition, the pangolin protein (TCF/LEF) is involved in the Wnt signaling pathway in pangolins, an intermediate host species for SARS (modified from [19, 51]).

### Physiological implications and importance of GSKIP in Wnt signaling

Our study revealed that the bat CLUH protein of composite-type GSKIP possesses a dominant consensus sequence (115SPxF118xxxF122xxxQ126xxRV130) quite similar to all the CLUH species described in the earlier text (S2 Table).

Overexpression of ATG2B and GSKIP increased progenitor sensitivity to thrombopoietin, enhancing megakaryocyte progenitor differentiation [11, 13]. The presence of the AK7 gene results in a predisposition to ciliary dyskinesia [43, 44, 49]. The protein encoded by ARMC4 containing 10 ARMs has also been reported to be involved in ciliary movement [43, 44].

The molecular mechanism for the coevolution of these neighboring genes with both simple-type (GSKIP-AK7) and composite-type GSKIP (ARMC4) in vertebrates remains unclear. A study suggested that pangolins are natural hosts of beta-coronaviruses, and comprehensive surveillance of coronaviruses in pangolins could improve our understanding of the spectrum of COVID-19 (the pandemic arising from a type of coronavirus called Sars-CoV-2; see [54] and S2 Table). Urgent investigation of the bat genome is required and may reveal the orchestration of specific elements in the COVID-19 crisis. ARMC4 may function as a competitor to

APC and β-catenin. In addition, the protein TCF4 in pangolins, as an intermediate host species of severe acute respiratory syndrome, indicates the involvement of the Wnt signaling pathway. This may be a coincidence but also raises the interesting question of what is the root cause of COVID-19 infection. Previously, we proposed that the ARMC4 family (as a composite-type GSKIP with a physiological role) together with simple-type GSKIP could function as competitors and is involved in the Wnt signaling pathway [20]. Both simple-type GSKIP and composite-type GSKIP (ARMC4) have been implicated in the Wnt signaling pathway (Fig 6) [50, 51, 55, 56].

## Discussion

According to UniProt data, we found the domain/fusion mechanism of the CLU and ARMC4 families and the hijack (recombination) mechanism for two sporadic genes of Hsp70/DnaK [52, 53] and Mitofilin [46, 47] that were hijacked by the DUF727 domain. The evolution of these composite-type GSKIPs was also revealed by the gene location in the corresponding organelles.

In this study, we detected gene fusion events by using UniProt, T-Coffee, and ClustalW. A total of 1748 simple-type GSKIP proteins as components of gene fusion in composite-type GSKIP were also detected, many of which were from the CLUH and ARMC4 families. The predicted functional associations were with mitochondria and the Wnt signaling pathway. We demonstrated for the first time that gene fusion is a complex evolutionary process for CLUH and ARMC4, including phylogenetic distance. Gene fusion plays a key role in the evolution of gene architecture. We can observe its effect if gene fusion occurs in coding sequences [57]. When gene fusion occurs in the assembly of a new gene, new functions emerge through the addition of peptide modules to the multidomain protein.

The detection methods for gene fusion events on a large biological scale can provide insights into the multimodular architecture of proteins [26, 58, 59].

In bacteria, a novel ligand-binding domain (ATP-binding motif), named the ACT domain, was identified through a PSI-BLAST search. *E. coli* 3PGDH was the first protein with an ACT domain that was found to fold into a ferredoxin-like (potential ATP in photosynthetic bacteria) βαββαβ topology. The ACT domain is found in a variety of contexts and is proposed to be a conserved regulatory ligand-binding fold. However, DUF727 has an antiparallel ββββ topology (NMR PDF file), sharing a similar sheet homology (Fig 1D and [2]) as the human CLUH--KIAA0664-SSF103107 superfamily domain (Fig 1E). Thus, Hsp70/Dnak, which requires ATP, chaperones, mitochondria, and Mitofilin, requires an ATP source for binding to CLU1. In particular, this standalone ACT domain protein might form complexes upon binding to other proteins, such as kinases, which interact with and regulate ARMC4, β-catenin, and APC in the Wnt pathway. We also observed that DUF727 fused to CLU1 and ARMC4. Moreover, domain accretion through a gene fusion mechanism may be a major contributor to gene evolution [60].

Three GSKIP-containing isoforms in humans are located in different chromosomes: GSKIP in chromosome 14, CLUH/KIAA0664 in chromosome 17, and ARMC4 in chromosome 10, and GSKIPs, CLUH, and ARMC4 still retain a DUF727. This may elucidate certain events in the evolutionary process. In particular, the unique order of Hsp70/Dnak and Mitofilin evolution may be evidence of another mechanism (hijacking). Our queries resulted in several new findings. First, we found a highly primitive form of GSKIP in lower organisms, such as gram-positive bacteria, and neither a PKA-binding domain nor a GSK3β-binding site was found when tracing GSKIP homologs in vertebrates and invertebrates [27, 37]. Second, the DUF727 domain in CLU1 and ARMC4, regarded as the central fragment of GSKIP, can exist

alone or with other domains as part of multidomain structures. Third, DUF727 first extended to a pre-GSK3β site at 115SPXF118 in vertebrate ARMC4 during evolution. Fourth, two sporadic composite-type GSKIPs (Mitofilin and Hsp70/Dnak) resulting from a hijacking mechanism were found in invertebrate nematodes. As shown in Fig 1F, at an evolutionarily later time, simple-type GSKIP acquired the PKA RII-binding domain as GSKIP/AKAP in vertebrates. Simple-type GSKIP is still retained in invertebrates. For composite-type GSKIP, one bacterium containing DUF727 with the ACT domain was found, whereas in all eukaryotes, DUF727 was improperly recognized as GSKIP. Although DUF727 lacks the ideal GSK3β-binding domain, it is still counted in CLU/TIF31 and CLUH/KIAA0664 sequences. In some vertebrates, the GSKIP domain was found in an ARMC4-containing protein with a pre-GSK3β-binding site only. The two sporadic proteins of Mitofilin and Hsp/DnaK were found in invertebrates containing perfect pre-GSK3β-binding and GSK3β-binding sites.

In sum, this study generated evidence demonstrating that the composite-type GSKIP of CLU1 and Mitofilin in mitochondria and the Hsp70 chaperone and ARMC4 have functions in Wnt signaling and simple-type GSKIP-linked proteins have roles in the origin of GSK3β-binding sites during DUF727/GSKIP evolution. Both simple-type and composite-type GSKIP (ARMC4) attract the most attention due to their degree of involvement in the Wnt signaling pathway (Fig 1F and Fig 6). Moreover, the entire genome of *Rhinolophus ferrumequinum* (the greater horseshoe bat) should be sequenced because doing so can provide insights into the importance of the bat genome. The likely insights into the bat genome might contribute to the efforts to end the current COVID-19 outbreak in so far as the insights may elucidate how GSKIPs function in bat mitochondria and Wnt signaling pathways; particularly in immunology studies may also help to understand how mother nature of bat genome harboring coronavirus could be used to combat emerging variants of this pandemic virus.

Two questions remain unanswered: first, under what conditions did this recombination occur to form composite-type GSKIP (to mitochondria, chaperone proteins, or armadillo repeats)? Second, in which organism did this recombination occur? For evolutionary biologists, these composite-GSKIP proteins can reveal the key steps in the evolution of GSKIP, particularly for composite-type GSKIP.

## Conclusions

Composite-type GSKIPs demonstrated the coevolution of pre-GSK3β- and GSK3β-binding sites that extended to DUF727 in the CLU1 and ARMC4 families, and the study findings may provide insights into the importance of both simple-type and composite-type GSKIP for the mitochondrial and Wnt signaling pathways.

## Supporting information

**S1 Table. Quote terms: SSF103107 (T3), PF05303 (T1), IPR007967 (G1), and IPR023231 (G3) of all species in the UniProt search.** T: type, G: group.
(DOCX)

**S2 Table. Quote terms: "Clustered mitochondria protein" (CLUH) of Bat in UniProt search.**
(DOCX)

**S1 Fig. Phylogenetic analysis of composite-type GSKIP, CLU1, and ARMC4.** Sequence alignment and phylogenetic analysis of ARMC4 from 18 vertebrate species along with a bacterium are shown. A and B, Phylogenetic tree of CLU1 and ARMC4 generated using MEGA-X, respectively. Boot strapping values represented the likelihood function for each other species

as indicated. The boot strapping test measures the internal consistency of data produced above than 0.5 (50%) of the bootstrap replicates are consistent. C, Multiple sequence alignment with respect to the DUF727 region of GSKIP orthologs was conducted using ClustalW. D, T-Coffee estimates of alignment accuracy improved phylogenetic tree reconstruction. The conserved residues are indicated with asterisks, and residues with high similarity among the orthologs are marked with dots at the bottom. The label * indicates the possible region of DUF727 in ARMC4.
(TIF)

## Acknowledgments

We thank Gary Mawyer and Wallace Academic Editing for English edition on the manuscript.

## Author Contributions

**Conceptualization:** Cheng-Yu Tsai, Shean-Jaw Chiou, Huey-Jiun Ko, Chen-Yen Lin, Jiin-Tsuey Cheng, Joon-Khim Loh.

**Data curation:** Huey-Jiun Ko, Aij-Li Kwan.

**Formal analysis:** Chen-Yen Lin, Chihuei Wang.

**Project administration:** Yi-Ren Hong.

**Software:** Yu-Fan Cheng, Sin-Yi Lin, Yun-Ling Lai, Chen-Yen Lin, Chihuei Wang, Hsin-Fu Liu.

**Validation:** Hsin-Fu Liu.

**Visualization:** Sin-Yi Lin.

**Writing – original draft:** Cheng-Yu Tsai, Huey-Jiun Ko, Chen-Yen Lin.

**Writing – review & editing:** Cheng-Yu Tsai, Shean-Jaw Chiou, Huey-Jiun Ko, Yu-Fan Cheng, Yun-Ling Lai, Chen-Yen Lin, Jiin-Tsuey Cheng, Aij-Li Kwan, Joon-Khim Loh.

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
