## [Decision Letter · Decision Letter 0]

22 Jul 2021

PONE-D-21-20264

Deciphering the evolution of composite-type GSKIP in mitochondria and Wnt signaling pathways

PLOS ONE

Dear Dr. Hong,

Thank you for submitting your manuscript to PLOS ONE. After careful consideration, we feel that it has merit but does not fully meet PLOS ONE’s publication criteria as it currently stands. Therefore, we invite you to submit a revised version of the manuscript that comprehensively addresses the points raised during the review process.

We look forward to receiving your revised manuscript.

Kind regards,

Michael Schubert

Academic Editor

PLOS ONE

Reviewers' comments:

Reviewer's Responses to Questions

**Comments to the Author**

1. Is the manuscript technically sound, and do the data support the conclusions?

Reviewer #1: Yes

Reviewer #2: Yes

Reviewer #3: Partly

2. Has the statistical analysis been performed appropriately and rigorously? 

Reviewer #1: I Don't Know

Reviewer #2: N/A

Reviewer #3: N/A

3. Have the authors made all data underlying the findings in their manuscript fully available?

Reviewer #1: Yes

Reviewer #2: Yes

Reviewer #3: Yes

4. Is the manuscript presented in an intelligible fashion and written in standard English?

Reviewer #1: No

Reviewer #2: No

Reviewer #3: No

5. Review Comments to the Author

Reviewer #1: GSKIP paper

The authors provide information about the evolution of GSKIP. The bioinformatics used is probably suitable for the analysis of evolutionary events. However, to this reviewer who is not a specialist in evolution and analysis of evolution, it is almost impossible to follow the arguments throughout the manuscript.

The authors need to define certain technical terms, most important what they mean by simple and composite types. What is meant by pre-GSK3beta?

The introduction nicely introduces GSKIP. However, then in lines 97-99 the link to context-dependency and Wnt signalling is completely obscure. Please explain better.

Fig. 1 provides structural information. It would be most valuable to overlay the different structure to visualize overlap/differences.

The authors suggest throughtout the manuscript that GSKIP gained the AKAP function in vertebrates. However, Hundsrucker et al, JBC 2010, had shown that GSKIP of C. elegans already binds PKA R subunits. Please address the discrepancy.

Line 405 sentence starting “In addition, Mitofilin…” is unconnected to the preceeding text. Please embed better.

Please do not term mutant versions of any binding domains or proteins “flawed”, please use mutant.

Starting with line 437, Hundsrucker had already addressed the question of whether lower GSKIP of lower organisms binds GSK3beta (Hundsrucker JBC 2010). Please consider in your analysis.

Line 462, what is meant by dominant consensus sequence?

Line 472, the link to Covid is unclear, please clarify and explain better.

Fig. 6, the title is misleading since – to the understanding of this reviewer- Dema et al., JBC 2016, had clearly shown a role of GSKIP in Wnt signalling.

Line 527, discussion, the paragraph is unconnected to the previous discussion. Please provide clear links.

Similary, line 562, the paragraph about Covid. What is the reason for speculating on Covid and GSKIP. Please substantiate.

In the conclusion, line 577, the thoughts about target-based drug discovery are not at all embedded into the discussion.

Reviewer #2: Tsai et al. investigated the role of Glycogen synthase kinase interaction protein (GSKIP) in mitochondria and the Wnt signaling pathways.

The manuscript needs to be re-written. Both, English must be improved including the structure of most sentences and the thread, which guides the reader through the information given by the authors.

The introduction is a numeration of findings. It is not clear, how the findings from the introduction should guide to the research question answered by the authors.

Methods: The authors mention a few times, that information were given previously, without giving a reference. Information about the used material is incomplete (e.g. line 173; 174; 179; …).

Into the discussion, there is redundancy of the results, e.g. from page 33.

It is not clear, why the authors stat that the yawn to sequence the whole genome of Rhinolophus ferrumequinum. Please, explain in more detail. However, such ideas are not sufficient to be present in a discussion. In addition, the link from the finding presented here, to Sars-CoV-2 (The virus is called Sars-CoV-2 and the disease arising from an infection is called COVID19) only can be assumed, but it is not clearly stated by the authors.

The conclusion is not clear. Please, only conclude from your results and do not speculate bats and viruses. The results do not support this.

Reviewer #3: The authors set the foundations of some interesting thoughts and ideas in their manuscript, but those ideas are clouded by poor judgment of their results and a lack of clarity in the writing that weighs the manuscript down. For example, the authors wrote (line 272) “to determine the conserved sequence of GSK3β-binding regions and DUF727 in CLU1, we selected one bacterial prototype and 20 GSKIP orthologs and used ClustalW and MEGA-X to construct a phylogenetic tree”. However, a phylogenetic tree is not the best option to determine conservation between sequences, a sequence alignment would be a more adequate choice. Moreover, the values represented in the phylogenetic tree (Fig. 2), if they mean bootstrap, are too low to be considered significant (e.g., 0.37; 0.25). The authors should include more information that clarifies the meaning of those values and reconsider if the phylogenetic tree is relevant. Secondly, the authors used the yeast two-hybrid system to investigate different GSK3β-binding sites and claim several positions as crucial for the binding when they use a fragment of the protein, however, the authors overlook the results when they use the entire protein. Thirdly, the manuscript suggests a certain anthropocentric tone with concepts like lower and higher animals. I recommend the authors to read the following short paper (https://doi.org/10.1002/bies.201370093) and I encourage them to reconsider the use of lower and higher animals.

Regarding the lack of clarity of the manuscript, I would like to point out that it is easy to find sentences out of context (e.g. line 139: “It can occur as a result of translocation, interstitial deletion, or chromosomal inversion.”; line 304; “Its inactivation with GSK3β was found when this site was modified.”), without enough clarity (e.g. line 132: “Accession number (S1 Table) and amino acid sequence (S1 Fig) were used as parameters.”; line:359: “The protein encoded by ARMC4, 10 armadillo repeat motifs (ARMs), and one HEAT repeat were contained in ARMC4, which is thought to be involved in ciliary and flagellar movement”), or hardly readable (e.g. line 542: “Second, in CLU1 and ARMC4, the DUF 727 domain, only as central fragment of GSKIP, can exist alone or with other domains as part of multi-domain structures.”) among others. Moreover, several statements lack a reference (e.g., line 125 after “gene products”; line 143 after “previously described”; line 169 “previously described”; line 207 after “Armadillo repeats”; line 304 after “humans”) or some references are not the correct ones (e.g., line 209; line 211; line 295; line 367; line 369). Authors should check that no error has occurred during the reference formatting process. Considering previous publications, from the same authors, this manuscript is not as clear as the others. The authors should give more context in the abstract and introduction to avoid confusion. Besides, a scheme of the workflow that they employed in their domain survey will help the reader follow the process. In relation to that, the authors need to make the results section “Analysis of UniProt and GSKIP gene ontology: Identification of composite-type GSKIP” more straightforward. The authors could, for example, refer more to their own figures (i.e., adding Fig. 1G in line 207 after “Armadillo repeats”; adding Fig. 1G in line 229 after GSKIP; adding Fig. 1F in line 236 after “invertebrates”) and maybe represent figures 1A, B and C as Venn diagrams. In line 253, the authors say “Using GSKIP as a keyword, a total of 1849 protein entries were shown” and in lines 255 authors say “Using CLU1 as a keyword, 879 protein entries were shown”, but the figure shows 2234 entries for GSKIP and 936 for CLU1. The authors need to check why those are different. Furthermore, in the results section “Two groups of composite-type GSKIP families may have evolved through a gene-fusion mechanism: CLU1 mitochondria) and ARM4 (Wnt pathway)” the list of accession numbers (lines 276 to 288) does not help the readability of the manuscript. Authors could add those numbers in figure 2 or in the supplemental material. Also, the authors should be more precise when they name the groups for the different species (i.e., line 280 “worm” is very unspecific, authors should consider “round-worm” or "nematode"; in lines 284 and 285, the authors wrote that the phylogenetic tree have one chordate and six vertebrates, however, vertebrates are chordates too, so the phylogenetic tree has seven chordates among them one cephalochordate [Branchiostoma lanceolatum] and six vertebrates.

Finally, some typographical errors that should be corrected:

Line 100: add “that” before co-exist

Line 116 and line 117: redundant reference, also this is not the format to cite Uniprot. The correct cite is https://doi.org/10.1093/nar/gkaa1100

Figure 1E: In the secondary structure drawing the letters are cut

Line 277: “Schizosac charomyces” should be one word

Line 289: “S3 Fig” should be “Fig 3”

Line 339: “light blue” should be “green”

Line 371 and line 373: “MARC4” should be “ARMC4”

6. PLOS authors have the option to publish the peer review history of their article (what does this mean?). If published, this will include your full peer review and any attached files.

Reviewer #1: No

Reviewer #2: No

Reviewer #3: **Yes: **Josep Marti-Solans

---

## [Author Response · Author response to Decision Letter 0]

5 Sep 2021

Reviewer #1: GSKIP paper

Q1: The authors provide information about the evolution of GSKIP. The bioinformatics used is probably suitable for the analysis of evolutionary events. However, to this reviewer who is not a specialist in evolution and analysis of evolution, it is almost impossible to follow the arguments throughout the manuscript.

A1: We thank the reviewer for your time and consideration in reviewing our article. Your constructive comments are encouraging, for which we are very grateful. We have addressed the comments as thoroughly as possible in the revised version of the manuscript. In particular, we have rewritten our abstract and introduction to improve our rationales.

Q2: The authors need to define certain technical terms, most important what they mean by simple and composite types. What is meant by pre-GSK3beta?

A2: Thank you for your comment. For the simple-type GKSIP, please see C.Hundsrucker, et al. JBC. 2010, figure 5. For the composite-type GSKIP, refer to Chou CH, et al. BBA. 2018, figure 2C. Both types of GSKIP have been described in the text.

For the mean of pre-GSK3beta binding site, during the evolution process of DUF727/GSKIP, the pre-GSK3beta binding site is prior to the GSK3beta binding site. However, it is particularly for the evolutionarily conserved pre-GSK3 binding site (115SPxF118) of composite-type GSKIP in both vertebrate and invertebrate, compared to evolved simple-type GSKIP (most of the species are 115SPxY118).

Q3: The introduction nicely introduces GSKIP. However, then in lines 97-99 the link to context-dependency and Wnt signalling is completely obscure. Please explain better.

A3: Thanks for your comment. We have revised those sentences in lines 93-96.

Q4: Fig. 1 provides structural information. It would be most valuable to overlay the different structure to visualize overlap/differences.

The authors suggest throughtout the manuscript that GSKIP gained the AKAP function in vertebrates. However, Hundsrucker et al, JBC 2010, had shown that GSKIP of C. elegans already binds PKA R subunits. Please address the discrepancy.

A4: Thank you for your comment. Hundsrucker et al, JBC 2010, indicated that the PKA binding site only exists in simple-type GSKIP as an AKAP, but not in composite-type GSKIP. In this manuscript, all the composite-type GSKIP (CLU1 and ARMC4) lack the PKA binding site, thus, the composite-type GSKIP cannot be recognized as an AKAP. 

Q5: Line 405 sentence starting “In addition, Mitofilin…” is unconnected to the preceeding text. Please embed better.

A5: Thanks for your suggestion. We have already revised this paragraph (lines 406-412).

Q6: Please do not term mutant versions of any binding domains or proteins “flawed”, please use mutant.

A6: Thank you for your suggestion. We have replaced flawed to mutant.

Q7: Starting with line 437, Hundsrucker had already addressed the question of whether lower GSKIP of lower organisms binds GSK3beta (Hundsrucker JBC 2010). Please consider in your analysis.

A7: In Hundsrucker’s study (JBC 2010), they only showed that the simple-type GSKIP of lower organisms binds GSK3beta.

In our study, we performed mutagenesis assay of various mutant GSK3β-binding sites from both invertebrate (lower organisms) and vertebrate (higher organisms) of composite-type CLU1, particularly for the pre-GSK3β-binding site (118F or 118Y).

Q8: Line 462, what is meant by dominant consensus sequence?

A8: Dominant consensus sequence referring to 115SPxF118xxxF122xxxQ126xxRV130 was found in composite-type of CLU1, which exists in many species, particularly for several sites at F118, Q126, V130 are not perfect compared to the ideal consensus sequence (115SPxY118xxxF122xxxL126xxRL130) for simple-type GSK3beta binding site (Please see text in lines 244, 553).

Q9: Line 472, the link to Covid is unclear, please clarify and explain better.

A9: Indeed, we have already discussed your concern in detail for COVID-19 and possible ARMC4 in Wnt signaling pathway in our result section (lines 458-477). Moreover, we expect to obtain the available entire genome sequence of Rhinolophus ferrumequinum (the Greater horseshoe bat) because it can provide insights into the importance of the bat genome (lines 558-566). This might contribute to the efforts of the current COVID-19 outbreak for guiding GSKIPs in bat mitochondria and signaling pathways along with immunology (how mother nature of bat genome harboring coronavirus) to combat emerging pandemic viruses. 

Q10: Fig. 6, the title is misleading since – to the understanding of this reviewer- Dema et al., JBC 2016, had clearly shown a role of GSKIP in Wnt signalling.

A10: We agreed. Dema et al., JBC 2016 and Chou CH, et al. BBA. 2018 only addressed the role of simple-type GSKIP in Wnt signalling.

In this study, we found both simple-type and composite-type in the title of fig 6. Particularly, we uncovered and emphasised that ARMC4 (composite-type GSKIP) and pangolin (beta-catenin, armadillo domain) might also be involved in the Wnt signaling pathway.

Q11: Line 527, discussion, the paragraph is unconnected to the previous discussion. Please provide clear links.

A11: Thank you for your suggestion. We have eliminated the unconnected sentence and linked the part of ATP (lines 523-524).

Q12: Similary, line 562, the paragraph about Covid. What is the reason for speculating on Covid and GSKIP. Please substantiate.

A12: Thank you for your comment. Please see our answer 9. From lines 466 to 481 and figure 6 “Physiological implications and importance of both simple-type and composite-type GSKIP in Wnt signaling”, we have already provided many concepts for COVID and GSKIPs. Furthermore, the genome of the bat (Rhinolophus ferrumequinum, the Greater horseshoe bat), which might be the possible source of COVID, has not been well-established yet. It can provide insights into the importance of the simple-type GSKIP in the bat’s genome. The likely insights into the bat genome might contribute to the efforts to end the current COVID-19 outbreak in so far as the insights may elucidate how GSKIPs function in bat mitochondria and Wnt signaling pathways; particularly in immunology studies may also help to understand how mother nature of bat genome harboring coronavirus could be used to combat emerging variants of this pandemic virus. (lines 558-566)

Q13: In the conclusion, line 577, the thoughts about target-based drug discovery are not at all embedded into the discussion.

A13: We agree. We have modified that paragraph in the conclusion.

Reviewer #2: Tsai et al. investigated the role of Glycogen synthase kinase interaction protein (GSKIP) in mitochondria and the Wnt signaling pathways.

Q1: The manuscript needs to be rewritten. Both, English must be improved including the structure of most sentences and the thread, which guides the reader through the information given by the authors.

A1: Your comment is well-taken. We have already rewritten the whole manuscript. Also, we have addressed the comments as thoroughly as possible in the revised version of the manuscript. 

Q2: The introduction is a numeration of findings. It is not clear, how the findings from the introduction should guide to the research question answered by the authors.

A2: Thanks for your suggestions. We have rewritten the rationale of your concerns in lines 93-96 “Collectively, simple-type GSKIP was believed to play a multifaceted role in several diseases oriented and a context-dependent manner, involved in mitochondria and the regulation of Wnt signaling pathway”. Based on above finding, we further explored the evolution of composite-type GSKIP (CLU1 and ARMC4) involving in mitochondria and the regulation of Wnt signaling pathway.

Q3: Methods: The authors mention a few times, that information were given previously, without giving a reference. Information about the used material is incomplete (e.g. line 173; 174; 179; …).

A3: Thank you for your comment. We have added references in line 166.

Q4: Into the discussion, there is redundancy of the results, e.g. from page 33. It is not clear, why the authors stat that the yawn to sequence the whole genome of Rhinolophus ferrumequinum. Please, explain in more detail. However, such ideas are not sufficient to be present in a discussion. 

A4: In fact, we have already discussed your concern in detail for COVID-19 and possible ARMC4 in Wnt signaling pathway in our result section (lines 468-481) “A study suggested that pangolins are natural hosts of beta-coronaviruses, and comprehensive surveillance of coronaviruses in pangolins could improve our understanding of the spectrum of COVID-19…”

To avoid the redundancy of the results and discussion here, we have modified the paragraph from lines 558-566. In this study, we are eagerly waiting for the whole genome sequence of Rhinolophus ferrumequinum. While the GSKIPs (simple-type and composite-type: Clu1; ARMC4; pangolin as a protein or intermediated host?) are somehow feeble, and the mechanisms are yet to be proved.

Q5: In addition, the link from the finding presented here, to Sars-CoV-2 (The virus is called Sars-CoV-2 and the disease arising from an infection is called COVID19) only can be assumed, but it is not clearly stated by the authors.

A5: We agree. We have added the sentence describing COVID-19 in lines 470-471 and cited in the references 54.

Q6: The conclusion is not clear. Please, only conclude from your results and do not speculate bats and viruses. The results do not support this.

A6: We agree. We have eliminated the part of bat and COVID-19 in the conclusion section.

Reviewer #3: The authors set the foundations of some interesting thoughts and ideas in their manuscript, but those ideas are clouded by poor judgment of their results and a lack of clarity in the writing that weighs the manuscript down.

Q1: For example, the authors wrote (line 272) “to determine the conserved sequence of GSK3β-binding regions and DUF727 in CLU1, we selected one bacterial prototype and 20 GSKIP orthologs and used ClustalW and MEGA-X to construct a phylogenetic tree”. However, a phylogenetic tree is not the best option to determine conservation between sequences, a sequence alignment would be a more adequate choice. 

A1: We agree. In fact, our data of T-coffee have already shown a multiple sequence alignment with respect to the DUF727 region of CLU1 orthologues (also ARMC4). Additionally, T-Coffee in figure 3 was performed to estimate alignment accuracy and improve phylogenetic tree reconstruction. Therefore, we revised the content of line 272 that we “firstly” selected one bacterial prototype and 20 GSKIP orthologs and used ClustalW and MEGA-X to construct a phylogenetic tree….. and the sentence from lines 288-291” Both a GSK3β-binding motif (122Fxxx126LxxR/K/QL130)….”.

Q2: Moreover, the values represented in the phylogenetic tree (Fig. 2), if they mean bootstrap, are too low to be considered significant (e.g., 0.37; 0.25). The authors should include more information that clarifies the meaning of those values and reconsider if the phylogenetic tree is relevant. 

A2: Thanks for your comment, the low bootstrap values represented in the phylogenetic tree (e.g., 0.37; 0.25) could possibly due to several reasons: firstly, we mixed all the phylogenetic analysis of four composite-type GSKIP, including ARMC4, CLU1, HSP70/Dnak and Mitofilin, among different species (figure 2). Secondly, the generating of the phylogenetic tree is dependent on the sequence homology among different species, while the 3D structure of DUF727 (figure 1 D) folds into a central part of anti-parallel ββββ topology (figure 1), indicating that the similarity of the DUF727 structure among species. Finally, when we performed the bootstrap test in MEGA, we set the number of bootstrap replicates as 500 in CLU1 and ARMC4 analysis, respectively (Supplement figure 1A and B). As MEGA finishes the bootstrap test, the flap shows the bootstrap consensus tree, which is a consensus of the bootstrap replicate trees. The bootstrap test measures the internal consistency of data produced above than 0.5 (50%) of the bootstrap replicates are consistent.

Q3: Secondly, the authors used the yeast two-hybrid system to investigate different GSK3β-binding sites and claim several positions as crucial for the binding when they use a fragment of the protein, however, the authors overlook the results when they use the entire protein. 

A3: Your comment is well-taken. We used simple-type of GSKIP as an entire protein (1-139, full-length) GSKIP and GSKIPtide (115-139, fragment) as a template to perform mutagenesis assay of various flawed GSK3β-binding sites from different species of composite-type CLU1 particularly for pre-GSK3β-binding site (118F or 118Y). We also added the above paragraph to the text in the method section: Cloning, site-directed mutagenesis and DNA sequencing.

Q4: Thirdly, the manuscript suggests a certain anthropocentric tone with concepts like lower and higher animals. I recommend the authors to read the following short paper (https://doi.org/10.1002/bies.201370093) and I encourage them to reconsider the use of lower and higher animals.

A4: We agree. We eliminated all the terms of lower and higher in this article. Except for bacteria, we used “highly primitive form of GSKIP in lower organisms” in lines 535-536.

Q5: Regarding the lack of clarity of the manuscript, I would like to point out that it is easy to find sentences out of context (e.g. line 139: “It can occur as a result of translocation, interstitial deletion, or chromosomal inversion.”; line 304; “Its inactivation with GSK3β was found when this site was modified.”), without enough clarity (e.g. line 132: “Accession number (S1 Table) and amino acid sequence (S1 Fig) were used as parameters.”; line:359: “The protein encoded by ARMC4, 10 armadillo repeat motifs (ARMs), and one HEAT repeat were contained in ARMC4, which is thought to be involved in ciliary and flagellar movement”), or hardly readable (e.g. line 542: “Second, in CLU1 and ARMC4, the DUF 727 domain, only as central fragment of GSKIP, can exist alone or with other domains as part of multi-domain structures.”) among others. 

A5: Thanks for your comment. 

1. The sentence has been deleted.

2. The sentence in lines 302-303 had been modified to indicate that its inactivation with GSK3β when this site was modified.

3. We added more descriptions in lines 129-132 for the S1 table and S1 figure.

4. Please take a look of the following sentence (line 362) for ARMC4 “The term armadillo is derived from the historical name of the β-catenin gene in the fruit fly Drosophila, where the armadillo repeat was first discovered.”

5. We have modified lines 538-540.

Q6: Moreover, several statements lack a reference (e.g., line 125 after “gene products”; line 143 after “previously described”; line 169 “previously described”; line 207 after “Armadillo repeats”; line 304 after “humans”) or some references are not the correct ones (e.g., line 209; line 211; line 295; line 367; line 369). Authors should check that no error has occurred during the reference formatting process. 

A6: Thank you for your suggestions. We have already cited and checked all the references.

Q7: Considering previous publications, from the same authors, this manuscript is not as clear as the others. The authors should give more context in the abstract and introduction to avoid confusion.

A7: Thank you for your suggestions. We have rewritten our abstract and introduction, particularly for the evolutionarily conserved pre-GSK3 binding site (115SPxF118) of composite-type GSKIP in both vertebrate and invertebrate, compared to evolved simple-type GSKIP (most of the species are 115SPxY118).

Q8: Besides, a scheme of the workflow that they employed in their domain survey will help the reader follow the process. In relation to that, the authors need to make the results section “Analysis of UniProt and GSKIP gene ontology: Identification of composite-type GSKIP” more straightforward. The authors could, for example, refer more to their own figures (i.e., adding Fig. 1G in line 207 after “Armadillo repeats”; adding Fig. 1G in line 229 after GSKIP; adding Fig. 1F in line 236 after “invertebrates”) and maybe represent figures 1A, B and C as Venn diagrams. 

In line 253, the authors say “Using GSKIP as a keyword, a total of 1849 protein entries were shown” and in lines 255 authors say “Using CLU1 as a keyword, 879 protein entries were shown”, but the figure shows 2234 entries for GSKIP and 936 for CLU1. The authors need to check why those are different. 

A8: Thank you for your suggestions.

1. In fact, in figure 1G, we have already demonstrated all the domains of four groups of composite-type GSKIP, including AcuB, CLU1, ARMC4, and sporadic (Hsp70/Dnak and Mitofilin). To consider the result of searching for those evolved GSK3 binding sites and PKA binding sites in composite-type GSKIP of CLU1 and ARMC4 is zero, indicating PKA domain is not yet involved in the evolution. We therefore use the number but not Venn diagram to represent all the queries. 

2. It is our mistake. In line 252, the protein entries of GSKIP is not 1849, the correct one is 2234. In line 254, the protein entries of CLU1 is not 879, the correct one is 936. All the correct numbers have shown in lines 254-256.

Q9: Furthermore, in the results section “Two groups of composite-type GSKIP families may have evolved through a gene-fusion mechanism: CLU1 mitochondria) and ARM4 (Wnt pathway)” the list of accession numbers (lines 276 to 288) does not help the readability of the manuscript. 

A9: Thank you for your comment. In this study, we found interestingly that DUF727/GSKIP domain was combined with CLU1 gene (mitochondria) and ARMC4 gene (Wnt pathway), designated as composite-type GSKIP. Therefore, all the accession numbers represent and ensure these two groups of composite-type GSKIP families (lines 274-286). 

Q10: Authors could add those numbers in figure 2 or in the supplemental material. Also, the authors should be more precise when they name the groups for the different species (i.e., line 280 “worm” is very unspecific, authors should consider “round-worm” or "nematode"; in lines 284 and 285, the authors wrote that the phylogenetic tree have one chordate and six vertebrates, however, vertebrates are chordates too, so the phylogenetic tree has seven chordates among them one cephalochordate [Branchiostoma lanceolatum] and six vertebrates.

A10: Thank you for your suggestions. We have already corrected the name of the species and the number (in lines 279, 283). 

Q11: Finally, some typographical errors that should be corrected:

Line 100: add “that” before co-exist

Line 116 and line 117: redundant reference, also this is not the format to cite Uniprot. The correct cite is https://doi.org/10.1093/nar/gkaa1100

Figure 1E: In the secondary structure drawing the letters are cut

Line 277: “Schizosac charomyces” should be one word

Line 289: “S3 Fig” should be “Fig 3”

Line 339: “light blue” should be “green”

Line 371 and line 373: “MARC4” should be “ARMC4”

A11: Thanks for your time and kindly reminder. We have already corrected all the errors.

---

## [Decision Letter · Decision Letter 1]

6 Oct 2021

PONE-D-21-20264R1Deciphering the evolution of composite-type GSKIP in mitochondria and Wnt signaling pathwaysPLOS ONE

Dear Dr. Hong,

Thank you for submitting your manuscript to PLOS ONE. After careful consideration, we feel that it has merit but does not fully meet PLOS ONE’s publication criteria as it currently stands. Therefore, we invite you to submit a revised version of the manuscript that addresses the points raised during the review process. Note that, although the reviewers agree that the manuscript has been significantly improved, there are still a few important issues that need to be addressed before the manuscript is acceptable for publication.

We look forward to receiving your revised manuscript.

Kind regards,

Michael Schubert

Academic Editor

PLOS ONE

Reviewers' comments:

Reviewer's Responses to Questions

**Comments to the Author**

1. If the authors have adequately addressed your comments raised in a previous round of review and you feel that this manuscript is now acceptable for publication, you may indicate that here to bypass the “Comments to the Author” section, enter your conflict of interest statement in the “Confidential to Editor” section, and submit your "Accept" recommendation.

Reviewer #1: All comments have been addressed

Reviewer #2: All comments have been addressed

Reviewer #3: (No Response)

2. Is the manuscript technically sound, and do the data support the conclusions?

Reviewer #1: Yes

Reviewer #2: Yes

Reviewer #3: Partly

3. Has the statistical analysis been performed appropriately and rigorously? 

Reviewer #1: Yes

Reviewer #2: Yes

Reviewer #3: N/A

4. Have the authors made all data underlying the findings in their manuscript fully available?

Reviewer #1: Yes

Reviewer #2: Yes

Reviewer #3: Yes

5. Is the manuscript presented in an intelligible fashion and written in standard English?

Reviewer #1: No

Reviewer #2: Yes

Reviewer #3: No

6. Review Comments to the Author

Reviewer #1: Although the authors have addressed the concerns of this reviewer there are still a few things to clarify. It still has not been properly defined what is meant by simple-type and composite forms of GSKIP. In line, in the introduction the authors list all known functions of GSKIP but do not elaborate on which GSKIP form, simple-type or composite, can be allocated to which function. Please do so in order to put the reader into a position to follow.

Reviewer #2: All suggestions and comments were addressed by the authors. I have no further comment sor suggestions.

Reviewer #3: Deciphering the evolution of composite-type GSKIP in mitochondria and Wnt signaling pathways

A considerable improvement has already taken place in the comprehensibility of the manuscript, however, the manuscript is still difficult to follow and needs to rethink again. Besides, some issues should be addressed.

For example, the list of organisms from lines 275 to 287 is referring only to CLU1, but in the tree, there are more species. Maybe the authors would like to refer to Fig S1A instead of Fig 2 in line 274. Secondly, because the four composite-type GSKIP differ a lot in their domains the bootstraps in figure 2 are not be reliable. A bootstrap support above 95% is very good and very well accepted and a bootstrap support between 75% and 95% is reasonably good, anything below 75% is very poor support and anything below 50% is of no use. If authors want to show phylogenetic relationships between the four composite-type GSKIP I recommend them to use only the orthologous domains to construct the tree.

Another issue is that the authors claim in line 305 the mutation of Leu130 always causes abolition of the interaction, however in full-length GSKIP the interaction is positive for 3 out of 4 mutations tested.

7. PLOS authors have the option to publish the peer review history of their article (what does this mean?). If published, this will include your full peer review and any attached files.

Reviewer #1: No

Reviewer #2: No

Reviewer #3: **Yes: **Josep Marti-Solans

---

## [Author Response · Author response to Decision Letter 1]

25 Oct 2021

Reviewer #1: Although the authors have addressed the concerns of this reviewer there are still a few things to clarify.

Q1. It still has not been properly defined what is meant by simple-type and composite forms of GSKIP. In line, in the introduction the authors list all known functions of GSKIP but do not elaborate on which GSKIP form, simple-type or composite, can be allocated to which function. Please do so in order to put the reader into a position to follow.

A1. Sorry for the confusion of “simple-type and composite forms of GSKIP”. GSKIP has two subtypes, there is only one simple-type GSKIP, all the functions described in the introduction part belong to this type. However, there are four groups of composite-type GSKIP (clustered mitochondria protein 1 (CLU1) and Mitofilin in mitochondria, heat shock protein 70 (Hsp70), and Armadillo repeat-containing 4 (ARMC4). In this article, we try to decipher and emphasize the origin and evolution of these four groups of composite-type GSKIP (Referring lines 102-109). 

Reviewer #2: All suggestions and comments were addressed by the authors. I have no further comments or suggestions.

Thank you for all the suggestions and efforts.

Reviewer #3: Deciphering the evolution of composite-type GSKIP in mitochondria and Wnt signaling pathways A considerable improvement has already taken place in the comprehensibility of the manuscript, however, the manuscript is still difficult to follow and needs to rethink again. Besides, some issues should be addressed. 

Q1. the list of organisms from lines 275 to 287 is referring only to CLU1, but in the tree, there are more species. Maybe the authors would like to refer to Fig S1A instead of Fig 2 in line 274. 

A1. Thank you for your comment. We have referred to figure S1A in line 274.

Q2. Because the four composite-type GSKIP differ a lot in their domains the bootstraps in figure 2 are not be reliable. A bootstrap support above 95% is very good and very well accepted and a bootstrap support between 75% and 95% is reasonably good, anything below 75% is very poor support and anything below 50% is of no use. If authors want to show phylogenetic relationships between the four composite-type GSKIP I recommend them to use only the orthologous domains to construct the tree.

A2. We agree. In general, we believe that a reliable bootstrap between 75% and 95% is for individual orthologous domains to construct the tree. Our data is for four groups of orthologous domains (from human to bacteria should be more diverse) to construct the tree, therefore, a bootstrap 50% could be acceptable.

Q3. Another issue is that the authors claim in line 305 the mutation of Leu130 always causes abolition of the interaction, however in full-length GSKIP the interaction is positive for 3 out of 4 mutations tested.

A3. Thank you for your comment. Since Leu130 plays a crucial role in GSKIP binding to GSK3beta for simple-type GSKIP, our mutagenesis assay (table 1) of the full length of GSKIP (1-139) and GSKIPtide (115-139) 3 out of 4 mutations due to the conformational change (particularly for Pro130 causes distort helix in both full-length and GSKIPtide). (Please see references 1,3,6 and 20 in this study.)

---

## [Decision Letter · Decision Letter 2]

17 Dec 2021

Deciphering the evolution of composite-type GSKIP in mitochondria and Wnt signaling pathways

PONE-D-21-20264R2

Dear Dr. Hong,

We’re pleased to inform you that your manuscript has been judged scientifically suitable for publication and will be formally accepted for publication once it meets all outstanding technical requirements.

Kind regards,

Michael Schubert

Academic Editor

PLOS ONE

Reviewers' comments:

Reviewer's Responses to Questions

**Comments to the Author**

1. If the authors have adequately addressed your comments raised in a previous round of review and you feel that this manuscript is now acceptable for publication, you may indicate that here to bypass the “Comments to the Author” section, enter your conflict of interest statement in the “Confidential to Editor” section, and submit your "Accept" recommendation.

Reviewer #1: All comments have been addressed

Reviewer #2: All comments have been addressed

2. Is the manuscript technically sound, and do the data support the conclusions?

Reviewer #1: Yes

Reviewer #2: Yes

3. Has the statistical analysis been performed appropriately and rigorously? 

Reviewer #1: Yes

Reviewer #2: N/A

4. Have the authors made all data underlying the findings in their manuscript fully available?

Reviewer #1: Yes

Reviewer #2: Yes

5. Is the manuscript presented in an intelligible fashion and written in standard English?

Reviewer #1: Yes

Reviewer #2: Yes

6. Review Comments to the Author

Reviewer #1: The concerns of this reviewer have been addresses. This makes it easier to follow the paper. Thank you.

Reviewer #2: I have no further comments and suggestions. All questions ans issues have been rased by the authors.

7. PLOS authors have the option to publish the peer review history of their article (what does this mean?). If published, this will include your full peer review and any attached files.

Reviewer #1: No

Reviewer #2: No

---

## [Editor Report · Acceptance letter]

5 Jan 2022

PONE-D-21-20264R2 

Deciphering the evolution of composite-type GSKIP in mitochondria and Wnt signaling pathways 

Dear Dr. Hong:

I'm pleased to inform you that your manuscript has been deemed suitable for publication in PLOS ONE. Congratulations! Your manuscript is now with our production department. 

Kind regards, 

on behalf of

Dr. Michael Schubert 

Academic Editor

PLOS ONE